# Recruitment, consent and retention of participants in randomised controlled trials: a review of trials published in the National Institute for Health Research (NIHR) Journals Library (1997–2020)

Richard M Jacques, Rashida Ahmed, James Harper, Adya Ranjan, Isra Saeed, Rebecca M Simpson, Stephen J Walters

School of Health and Related Research, The University of Sheffield, Sheffield, UK

**Correspondence to**
Dr Richard M Jacques;
r.jacques@sheffield.ac.uk

## ABSTRACT

**Objectives** To review the consent, recruitment and retention rates for randomised controlled trials (RCTs) funded by the UK's National Institute for Health Research (NIHR) and published in the online NIHR Journals Library between January 1997 and December 2020.

**Design** Comprehensive review.

**Setting** RCTs funded by the NIHR and published in the NIHR Journals Library.

**Data extraction** Information relating to the trial characteristics, sample size, recruitment and retention.

**Primary and secondary outcome measures** The primary outcome was the recruitment rate (number of participants recruited per centre per month). Secondary outcomes were the target sample size and whether it was achieved; consent rates (percentage of eligible participants who consented and were randomised) and retention rates (percentage of randomised participants retained and assessed with valid primary outcome data).

**Results** This review identified 388 individual RCTs from 379 reports in the NIHR Journals Library. The final recruitment target sample size was achieved in 63% (245/388) of the RCTs. The original recruitment target was revised in 30% (118/388) of trials (downwards in 67% (79/118)). The median recruitment rate (participants per centre per month) was found to be 0.95 (IQR: 0.42–2.60); the median consent rate was 72% (IQR: 50%–88%) and the median retention rate was estimated at 88% (IQR: 80%–97%).

**Conclusions** There is considerable variation in the consent, recruitment and retention rates in publicly funded RCTs. Although the majority of (6 out of 10) trials in this review achieved their final target sample; 3 out of 10 trials revised their original target sample size (downwards in 7 out of 10 trials). Investigators should bear this in mind at the planning stage of their study and not be overly optimistic about their recruitment projections.

## INTRODUCTION

Randomised controlled trials (RCTs) are the 'gold-standard' research design for evaluating the effectiveness of interventions in

### Strengths and limitations of this study

► This is the largest comprehensive review of recruitment, consent and retention in trials to date reporting rates for 388 single and multicentre trials published in the National Institute for Health Research (NIHR) Journals Library between January 1997 and December 2020.

► As the NIHR Journals Library intends to publish all research from Efficacy and Mechanism Evaluation, Health Services and Delivery Research, Health Technology Assessment, Programme Grants for Applied Research and Public Health Research funded projects, this study has less chance of publication bias compared with a review of other journals where publishing is more selective.

► For some trials crude recruitment rates, assuming all centres were recruiting for the same time period, were calculated, these estimates may be an underestimation of the true recruitment rate.

► The review is restricted to publicly funded trials published in the NIHR Journals Library, which may limit the generalisability of the findings.

health, education and policy.[1] Conducting an RCT requires major financial investment and substantial amounts of public funding is spent in this area each year. In 2019/2020, the National Institute for Health Research (NIHR) in England awarded over £250 million of funding to 310 research projects with a substantial proportion of this invested in RCTs.[2]

There are many practical challenges associated with conducting clinical trials. The leading reason for premature discontinuation of RCTs is poor recruitment of participants[3 4] with accrual often taking longer or being more difficult than expected. Poor recruitment can have a number of consequences including the study being underpowered if the target



sample size is not met and increased costs if an extension is required.[5] Furthermore, discontinued RCTs are less likely to be published in medical journals[3] which has ethical implications around research waste.[6]

There have been a number of previous studies in the UK investigating recruitment and retention in publicly funded RCTs. The earliest review, a cohort of trials funded by the Medical Research Council and NIHR Health Technology Assessment (HTA) between 1994 and 2002, reported that 31% (38/122) of the trials successfully recruited to their original recruitment target, with 54% (65/122) of trials awarded a grant extension.[7] There is evidence of a marginal improvement in these figures over time, with results from a cohort of 151 RCTs funded by the NIHR HTA programme between 2004 and 2016 finding that 40% (61/151) of trials successfully recruited to their original sample size, and 32% (49/151) of trials extended their recruitment.[8] In the same study, the median recruitment rate was found to be 0.92 (IQR: 0.43–2.79) participants per centre per month.

Following the publication of the review by Walters et al[8] in 2017, there have been several Cochrane systematic reviews looking at strategies for improving recruitment[9] and retention[10] of participants in RCTs. Two strategies for improving recruitment were identified with high-certainty evidence: using open trials rather than blinded, placebo controlled trials, and telephone reminders to people who did not respond to postal invitations. There has also been a systematic review of statistical models for predicting recruitment at the design stage of a clinical trial[11] but a survey of statisticians in UK and European clinical trial networks found that 90% (62/69) did not use statistical models for recruitment prediction.[12] In 2014, a trials methodology research priority setting exercise was conducted using a Delphi survey of directors of UK Clinical Research Collaboration registered Clinical Trials Units (CTUs).[13] Two of the three highest priority areas were 'Research methods to boost recruitment in trials' and 'Methods to minimise attrition'.

The Consolidated Standards of Reporting Trials (CONSORT) statement was first published in 1996,[14] and revised in 2001[15] and 2010.[16] It is a checklist of standards for reporting how a trial was designed, analysed and interpreted, and it has been endorsed both by prominent general medical journals and many specialist medical journals.[17] However, reporting guidelines such as CONSORT are not adopted and adhered to as much as they should be[18] with the previous review of recruitment and retention in RCTs by Walters et al[8] finding that 63% (95/151) of trials demonstrated complete compliance with CONSORT statement and reported each of the number: screened, eligible, declined consent, recruited and assessed for their primary outcome.

This review aims to update previous research on how well recruitment and retention figures are reported, and the rates of recruitment and retention in trials published in the NIHR HTA Journal between January 2004 and April 2016.[8] In this study, we update and extend this review to look at trials published in the NIHR Journals library from January 1997 to December 2020.

## METHODS
### Trial identification
Reports of individually RCTs published in the NIHR Journals Library from January 1997 to December 2020 were reviewed. Established in 2006, the NIHR is now the largest funder of health and social research in England.[2] The NIHR Journals Library publishes five peer reviewed journals reporting the results from a range of health research areas: Efficacy and Mechanism Evaluation (EME), Health Services and Delivery Research (HS&DR), HTA, Programme Grants for Applied Research (PGfAR) and Public Health Research (PHR) (https://www.journalslibrary.nihr.ac.uk/journals/). The first volume of the HTA journal was published in 1997 whereas the other four journals are more recent with the first volumes of the HS&DR, PGfAR and PHR journals published in 2013 and the first volume of the EME journal published in 2014. Trial reports published in the NIHR Journals Library were chosen as they provide a detailed description of the research methods and study results including recruitment and retention information.

The reports for review were obtained from the NIHR Journals Library website (https://www.journalslibrary.nihr.ac.uk/—last accessed 10 November 2021) along with any published trial paper, protocol paper or trial protocol. The published International Standardised Randomised Controlled Trial Number (ISRCTN) was used where available to check the ISRCTN register of clinical trials for additional information (https://www.isrctn.com/). The titles and abstracts of all reports published in the five NIHR journals from 1 January 1997 to 31 December 2020 were checked for relevance.

### Inclusion/exclusion criteria
To ensure consistency the eligibility criteria used by Walters et al[8] was adopted. Reports included in the review were of single or multicentre RCTs that were either fully or partially randomised and where recruitment to the trial had finished. Reports of trials that terminated early, either prior to completion of recruitment or following recruitment but prior to completion of follow-up were retained. Reports of two or more parallel RCTs were included as were nested parallel trials as part of another RCT. Some reports in the PGfAR journal included multiple independent RCTs and each of these trials were included separately. Reports of non-RCTs, cluster RCTs, adaptive designs, influenza vaccination trials, follow-on studies and ongoing RCTs that had not completed recruitment were excluded. Reports of external pilot/feasibility studies were excluded as they do not contribute outcome data to the main trial and are instead often used to estimate parameters such as the number of eligible participants, willingness of participants to be randomised and follow-up rates needed for the design of the main

study.[19] Reports of internal pilot trials that either went on to contribute outcome data to a full trial or were terminated prior to the full trial because of recruitment issues were included in the review.

## Data extraction

After the NIHR reports had been selected for inclusion, information was extracted using a standardised data extraction form. For each of the included trials the following information was extracted.

► Trial characteristics, including the trial design, clinical area, type of intervention, type of control, number of arms, use of blinding of trial participant, geographical region, number of centres, any support provided by a CTU and whether there was any description of pilot or feasibility work done prior to the start of the trial.

► Sample size, recruitment and retention information, including the target and actual sample size, the overall and centre-specific recruitment period and CONSORT information on the numbers screened, consented, randomised and analysed for the primary outcome.[16]

The selection of RCTs and data extraction was conducted by a team of reviewers (RMJ, RA, JH, AR and IS). Three reviewers (RMJ, RMS and SJW) conducted quality assurance checks on 30% of the included trials after the data extraction was completed, and disagreements were discussed to achieve consensus.

## Analysis

The primary outcome for the review was the recruitment rate for each trial. This was defined as the number of participants recruited and randomised per centre per month. Where explicit dates were reported the recruitment rate was calculated as the time between the date of recruitment start and the date of recruitment completion. In cases where only the months of recruitment were reported the recruitment period was estimated as the time between the first of the month and the end of the final month. If the date of the first participant recruited was reported instead of the start date of recruitment then the start of recruitment was taken as the first of the month of the first participant recruited. When the start of recruitment was not reported the start of screening was used to calculate the recruitment period. The recruitment period was estimated by subtracting the length of the follow-up period from the length of the study period when explicit information on the start and end of recruitment was not reported.

The recruitment rate was calculated in two different ways. The overall recruitment rate was calculated as the total number of participants recruited divided by the maximum number of recruiting sites, then divided by the total number of months that the trial recruited for. This overall recruitment rate is likely to be an underestimate for multicentre trials because each trial site is unlikely to open for recruitment at the same time and will not recruit for the entire recruitment period. To allow for the difference in start-up times and recruitment periods between sites, where available, the site-specific recruitment periods were extracted. These were averaged over the number of sites to give an average site-specific recruitment period. The average recruitment rate was calculated as the total number of participants recruited divided by the maximum number of sites, then divided by the average number of months that the trial recruited for.

The secondary outcomes for the review were the target sample size and whether it was achieved, the consent rate and the retention rate. The consent rate was calculated as the percentage of eligible participants that consented and were randomised (ie, the total number of participants recruited and randomised divided by the number of eligible participants). The retention rate was calculated as the percentage of randomised participants that were assessed for the primary outcome and included in the analysis of the primary outcome (ie, the number of participants included in the analysis of the primary outcome divided by the number of participants recruited and randomised).

Recruitment rates were summarised using the median and IQR due to the skewed distribution of the data.[20] The median and IQR were also used to summarise the secondary outcomes of the consent and retention rates. Comparisons of recruitment and retention rates were made between different trial characteristics using appropriate non-parametric tests; Mann-Whitney U test (for characteristics with two levels), Kruskal-Wallis test (three or more nominal levels) and Jonckheere-Terpstra test (three or more ordered levels). Analysis was conducted on a complete case basis so where the characteristics information, recruitment rate or retention rate were missing these were excluded. All statistical analyses were conducted in R V.4.1.0,[21] figures were produced using the package ggplot2[22] and the Jonckheere-Terpstra test conducted using the package clinfun.[23]

## Patients and public involvement

Patients and/or the public were not involved in the design, conduct, reporting or dissemination plans of this research.

## RESULTS

Between 1 January 1997 and 31 December 2020, 1899 reports were published in the five NIHR journals. Following screening, 1299 of these were excluded as reports of non-RCTs. The search identified 600 reports of RCTs of which 221 were excluded after applying the exclusion criteria (101 cluster RCTs; 95 pilot/feasibility RCTs; 14 follow-on studies; 6 adaptive designs; 3 influenza vaccination trials and 2 ongoing trials). Eight NIHR reports described the results of multiple independent trials (7 reports described 2 RCTs and 1 described 3 RCTS), therefore in total, 388 individual RCTs from 379 reports were included in the review and analysed as shown in figure 1. This includes 151 RCTs from the review by Walters *et al*.[8]

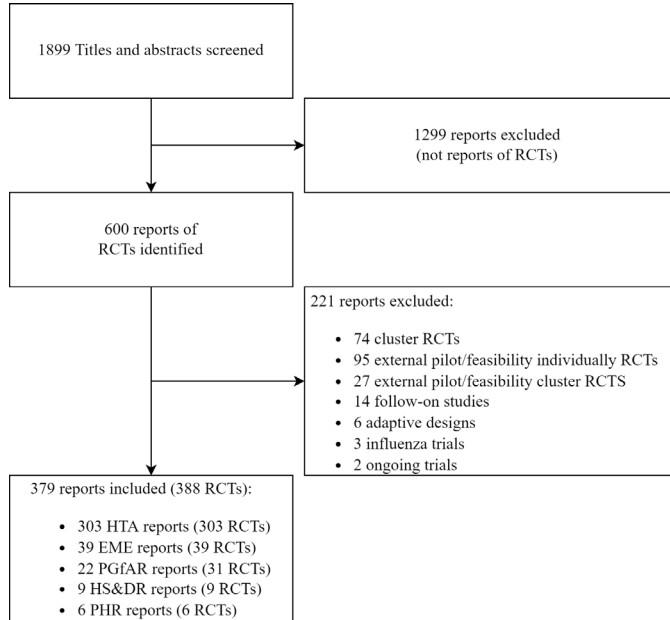

**Figure 1** Flow diagram of search and selection process of individually RCTs from the five NIHR journals between 1 January 1997 and 31 December 2020. EME, Efficacy and Mechanism Evaluation; HS&DR, Health Services and Delivery Research; HTA, Health Technology Assessment; NIHR, National Institute for Health Research; PGfAR, Programme Grants for Applied Research; PHR, Public Health Research; RCTs, randomised controlled trials.

## Trial characteristics

The characteristics of the 388 trials included in the review are summarised in table 1. The most common design was a two arm parallel group, multicentre RCT. The most frequently studied clinical areas were mental health, including psychiatry and psychology (19% (73/388) of trials) and musculoskeletal conditions, including orthopaedics, rheumatology and back pain (11% (44/388) of trials). The majority of trials were set in hospitals (56% (219/388)), took place in the UK (91% (355/388)) and across multiple geographical regions (82% (317/388)). Trials of pharmaceutical interventions (29% (111/388)) were more common than other interventions and 78% (301/388) of trials used an active control. Half of all trial reports (194/388) reported or mentioned work from a pilot or feasibility study.

The recruitment and sample size characteristics of the RCTs included in the review are summarised in table 2. The majority of trials (354/388) were multicentre with a median of 17 centres (IQR: 7–37). The final recruitment target (sample size) ranged from 44 participants to 46 000 participants and the final number recruited ranged from 2 participants to 47 062. The RCT with the highest final recruitment target and highest number recruited was an obstetrics trial investigating computerised interpretation of fetal heart rate during labour.[24] There were four trials that recruited less than 10 participants, two were discontinued at the end of an internal pilot phase due to low recruitment[25 26] and the remaining two had no pilot

| Table 1 | Characteristics of the trials included in the review | |
| --- | --- | --- |
| **Characteristic** | | **n (%)** |
| Trial design (n=388) | Parallel | 345 (89) |
| | Factorial | 19 (5) |
| | Crossover | 4 (1) |
| | Other* | 20 (5) |
| Arms (n=388) | 2 | 290 (75) |
| | 3 | 61 (16) |
| | 4 | 24 (6) |
| | >4 | 13 (3) |
| Clinical area (n=388) | Mental health | 73 (19) |
| | Musculoskeletal, orthopaedics and rheumatology | 44 (11) |
| | Obstetrics and gynaecology | 32 (8) |
| | Respiratory | 29 (7) |
| | Cardiovascular | 24 (6) |
| | Cancer/oncology | 21 (5) |
| | Stroke | 19 (5) |
| | Dermatology (including ulcers) | 17 (4) |
| | Gastrointestinal | 14 (4) |
| | Primary care | 11 (3) |
| | Diabetes | 11 (3) |
| | Urology | 10 (3) |
| | Neurology | 10 (3) |
| | Infectious disease | 8 (2) |
| | Dentistry | 5 (1) |
| | Other† | 60 (15) |
| Setting (n=388) | Hospital | 219 (56) |
| | General practice | 55 (14) |
| | Mixed | 61 (16) |
| | Community | 34 (9) |
| | Other‡ | 19 (5) |
| Intervention type (n=388) | Pharmaceutical intervention | 111 (29) |
| | Complex intervention | 65 (17) |
| | Therapy | 54 (14) |
| | Surgery | 46 (12) |
| | Other§ | 112 (29) |
| Control type (n=388) | Placebo | 87 (22) |
| | Active | 301 (78) |
| Patient blinded (n=384) | Yes | 100 (26) |
| | No | 284 (74) |
| Centres outside the UK? (n=388) | Yes | 33 (9) |
| | No | 355 (91) |
| | | Continued |

**Table 1** Continued

| Characteristic | | n (%) |
|---|---|---|
| Geographical spread (n=388) | Multiple regions | 317 (82) |
| | Regional | 71 (18) |
| Some form of pilot?¶ (n=388) | Yes | 194 (50) |
| | No | 194 (50) |

*Two or three parallel RCTs, cohort multiple RCT, patient preference/Zelen's.
†Alcohol abuse, allergy, chronic fatigue, cystic fibrosis, gerontology, hepatology, intensive care, minor surgery, multiple sclerosis, obesity/weight loss, nephrology, neurosurgery, nutrition, ophthalmology, otorhinolaryngology, paediatric (general, anaesthesiology, dermatology, nephrology, obesity/weight loss), physical exercise, rehabilitation, reproductive health resuscitation, septic shock, sleep disorders, speech therapy, vascular.
‡Bowel Cancer Screening Programme, Exercise Schemes, Football Clubs, HIV Clinics, Intellectual Disability Services, Leisure Centres, Mobile Dental Clinics, Online, Physical Therapy Classes, Prison, Public School, Sexual Health Clinics, Specialist Care Centres, Stop Smoking Services, University Clinics.
§Advice and Information, consultation, diagnostic Information, drug versus surgery, equipment, health professional, patient pathway, technique.
¶Any mention of pilot work or feasibility study recorded.
RCTs, randomised controlled trials.

phase.[27 28] Overall, 63% (245/388) of trials recruited to their final recruitment target but 32% (79/245) of these trials required an extension to their recruitment period to meet the target. A further 22% (86/388) of trials recruited to within 80% of their final recruitment target with 36% (31/86) of these trials having an extension to their recruitment period. The original recruitment target was revised in 30% (118/388) of trials (downwards in 67% (79/118)). For the majority of trials the primary outcome was collected at between 1 and 18 months postrandomisation.

### CONSORT and recruitment data

Summaries of the data completeness in relation to the CONSORT statement, recruitment and retention are presented in table 3. Of the 388 RCTs identified, 68% (265/388) fully complied with the CONSORT statement and reported the number of participants screened, eligible, declined consent, recruited and assessed for the primary outcome. The total number of participants recruited and randomised, and the number included in the analysis of the primary outcome, used to measure retention, was available for all 388 trials. Regarding the information required to calculate the recruitment rate, 98% (379/388) of trials reported the number of centres, 95% (369/388) reported the maximum length of the recruitment period, and 25% (97/388) reported the centre-specific recruitment information used to calculate an average recruitment period per centre. There was enough information reported to calculate the overall recruitment rate for 94% (365/388) of trials in this review.

### Recruitment, consent and retention rates

From the 365 trials with sufficient information to calculate the recruitment rate, the median was found to be 0.95 participants recruited per centre per month. The highest recruitment rate (57.75 participants per centre per month) was in a trial comparing medical to surgical termination of pregnancy with a target sample size of 2232 women[29] and the lowest (0.01 participants per centre per month) was in a trial treatment for transverse myelitis.[27] The 80th and 90th percentiles were found to be 3.70 and 9.47 participants recruited per centre per month, respectively. From the 22 single centre trials with sufficient information, the median recruitment rate was found to be 14.12 (IQR: 4.29–26.59, range: 1.58–57.75) participants per centre per month compared with a median of 0.86 (IQR: 0.40–2.17, range: 0.01–51.14) participant per centre per month in the 343 multicentre trials. Table 4 shows some statistical evidence of a difference in recruitment rates between the five NIHR journals (p=0.010) with the PHR journal having the highest median recruitment rate (7.62, IQR: 1.79–17.06) and the HTA journal having the lowest (0.85, IQR: 0.39–2.49). However, there are only six trials from the PHR journal included in this review and three of these trials[30–32] have a recruitment rate of 10 participants per centre per month or greater. Figure 2 shows the distribution of recruitment rates by clinical area. The highest median recruitment rate was for dentistry (1.95 participants recruited per centre per month) but this was only from five trials. The largest recruitment rates were found to be from four obstetrics and gynaecology trials,[24 29 33 34] a mental health trial[35] and three trials from other clinical areas.[36–38]

The median consent rate (percentage of eligible participants consented and randomised) was found to be 72% (IQR: 50%–88%). Table 4 shows some variability in consent rates between the journals with the HS&DR journal having the largest median rate (81%, IQR: 60%–97%) and the PHR journal the lowest (57%, IQR: 40%–68%). However, there is not an overall statistically significant difference in consent rates between the five NIHR journals (p=0.225). The median retention rate (per cent of randomised participants retained and assessed in the analysis of the primary outcome) was found to be 88% (IQR: 80%–97%). There were four trials[25 26 28 39] with a retention rate of 0%, these trials were all stopped early due to problems with recruitment and the planned statistical analysis for the primary outcome was not performed. Retention rates do not differ greatly between the five NIHR journals (p=0.118) (table 4).

The trial recruitment and retention rates are summarised by trial characteristics in tables 5 and 6, respectively. There is some statistical evidence of an association between the setting of the trial, final recruitment target and the total number of participants recruited but the median rates show no clear patterns to these associations.

The results of the current review, in terms of successful recruitment to target sample size, have been compared

**Table 2** Recruitment and sample size characteristics of the trials included in the review

| Characteristic (n=388) | | n (%) | Mean (SD) | Median (IQR) | Range |
|---|---|---|---|---|---|
| No of centres | 1 | 25 (6) | 29 (34) | 17 (7–37) | 1–274 |
| | 2–5 | 61 (16) | | | |
| | 6–10 | 48 (12) | | | |
| | 11–20 | 69 (18) | | | |
| | 21–50 | 112 (29) | | | |
| | 51–100 | 48 (12) | | | |
| | >100 | 16 (4) | | | |
| | Missing | 9 (2) | | | |
| Original target recruitment | ≤200 | 49 (13) | 1097 (3080) | 500 (300–900) | 50–46 000 |
| | 201–400 | 101 (26) | | | |
| | 401–600 | 86 (22) | | | |
| | 601–800 | 41 (11) | | | |
| | >800 | 109 (28) | | | |
| | Missing | 2 (1) | | | |
| Final target recruitment | ≤200 | 53 (14) | 1041 (3074) | 480 (270–802) | 44–46 000 |
| | 201–400 | 112 (29) | | | |
| | 401–600 | 84 (22) | | | |
| | 601–800 | 42 (11) | | | |
| | >800 | 97 (25) | | | |
| Final total recruitment | ≤200 | 72 (19) | 991 (3025) | 452 (236–800) | 2–47 062 |
| | 201–400 | 99 (26) | | | |
| | 401–600 | 82 (21) | | | |
| | 601–800 | 39 (10) | | | |
| | >800 | 96 (25) | | | |
| Final recruitment target achieved | Yes | 245 (63) | | | |
| | No, but with ≥80% of target | 86 (22) | | | |
| | No, <80% of target | 57 (15) | | | |
| Timing of primary outcome follow-up (months postrandomisation) | ≤1 month | 42 (11) | 12 (13) | 10 (3–12) | 0–120 |
| | 1< months ≤6 | 129 (33) | | | |
| | 6< months ≤18 | 131 (34) | | | |
| | >18 months | 63 (16) | | | |
| | Missing | 23 (6) | | | |
| Timing of final follow-up (months postrandomisation) | ≤1 month | 20 (5) | 16 (19) | 12 (6–18) | 0.066–144 |
| | 1< months ≤6 | 87 (22) | | | |
| | 6< months ≤18 | 181 (47) | | | |
| | >18 months | 88 (23) | | | |
| | Missing | 12 (3) | | | |

with three previous reviews[5 7 8] in table 7. As this review updates the findings of Walters et al[8] and due to there being some overlap with the trials included in Sully et al[5]; a column has been included for the non-overlapping time interval (2017–2020) in addition to the full time interval (1997–2020). Table 7 shows that 61% (107/174) of trials in the period 2017–2020 recruited 100% of the original target sample size which is higher than the previous periods/reviews. The target sample size was

revised in 31% (54/174) of trials; and the revision was downwards for 57% (31/54) of trials. An extension, to the trial timelines, was reported in 37% (65/174) of trials and this was higher than the review by Walters et al[8] (32% (49/151)) but lower than the reviews by McDonald et al[7] (54% (65/122)) and Sully et al[5] (45% (33/73)).

Figure 3 shows the percentage of trials recruiting 100% of the final target and 80% or more of the final target by publication year. There is no clear trend in the

**Table 3** Data completeness in relation to CONSORT guidelines and recruitment information

| Trial characteristic (N=388) | n (%) |
|---|---|
| No screened | 327 (84) |
| No eligible | 309 (80) |
| No refused/declined consent | 282 (73) |
| Total recruitment | 388 (100) |
| No included in primary analysis (retention) | 388 (100) |
| No of centres | 379 (98) |
| Maximum recruitment length | 369 (95) |
| Centre-specific recruitment length | 97 (25) |
| Recruitment rate can be calculated | 365 (94) |

CONSORT, Consolidated Standards of Reporting Trials.

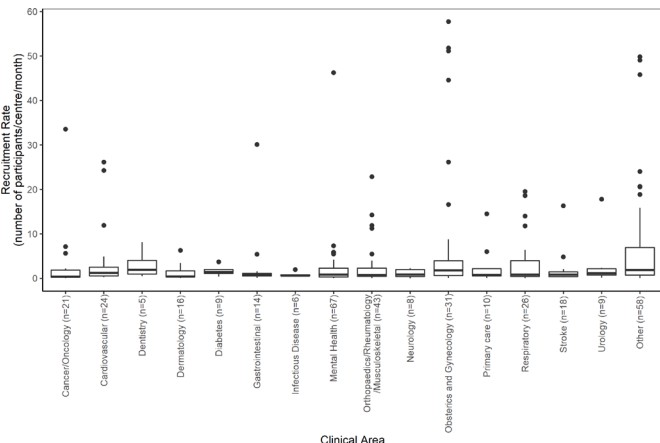

**Figure 2** Boxplots of recruitment rates by clinical area.

percentage of trials recruiting 100% of the final target for the earlier years (1999–2006) but there is evidence of an upward trend for the years 2007 to 2020.

## DISCUSSION AND CONCLUSIONS

This study has systematically conducted a review of the recruitment and retention data from a cohort of 388 trials published in the NIHR Journals Library between 1997 and 2020. This review found that the final target sample size was achieved in 63% (245/388) of RCTs; the median

recruitment rate was 0.95 (IQR: 0.42–2.60) participants per centre per month; the median consent rate was 72% (IQR: 50%–88%); and the median retention rate was 88% (IQR: 80%–97%).

This review found that 53% (207/388) of publicly funded RCTs achieved their original target sample size. Restricting the time period to 2017–2020 the figure is 61% (107/174), this is higher than the previous figures of 55% and 40% found in the reviews by Sully *et al*[5] and Walters *et al*.[8] This is also reflected in the percentage of trials recruiting to 100% of their final target where there is some evidence of an upward trend for the years 2007

**Table 4** Overall consent, recruitment and retention rates and association with Journal

| | Journal | n | Median | IQR | Range | P value |
|---|---|---|---|---|---|---|
| Consent rate (percentage of eligible participants consented and randomised) | All | 309 | 72% | 50%–88% | 4%–100% | |
| | HTA | 230 | 72% | 50%–88% | 4%–100% | 0.225* |
| | EME | 36 | 74% | 52%–93% | 11%–100% | |
| | PGfAR | 30 | 65% | 48%–84% | 19%–100% | |
| | HS&DR | 7 | 81% | 60%–97% | 35%–100% | |
| | PHR | 6 | 57% | 40%–68% | 35%–76% | |
| Recruitment rate (participants recruited per centre per month) | All | 365 | 0.95 | 0.42–2.60 | 0.01–57.75 | |
| | HTA | 289 | 0.85 | 0.39–2.49 | 0.01–57.75 | 0.010* |
| | EME | 39 | 1.18 | 0.45–2.46 | 0.15–18.61 | |
| | PGfAR | 25 | 1.18 | 0.53–2.80 | 0.07–24.03 | |
| | HS&DR | 6 | 1.88 | 1.71–10.82 | 1.69–18.87 | |
| | PHR | 6 | 7.62 | 1.79–17.06 | 1.69–20.57 | |
| Retention rate (percentage of randomised participants retained and assessed in primary outcome) | All | 388 | 88% | 80%–97% | 0%–100% | |
| | HTA | 303 | 89% | 80%–97% | 0%–100% | 0.118* |
| | EME | 39 | 89% | 80%–97% | 47%–100% | |
| | PGfAR | 31 | 84% | 78%–91% | 43%–100% | |
| | HS&DR | 9 | 82% | 73%–89% | 68%–99% | |
| | PHR | 6 | 85% | 78%–90% | 74%–92% | |

*P values are reported from a Kruskal-Wallis test.
EME, efficacy and mechanism evaluation; HS&DR, Health Services and Delivery Research; HTA, Health Technology Assessment; PGfAR, Programme Grants for Applied Research; PHR, Public Health Research.

**Table 5** Association between recruitment rate (number of participants/centre/month) and trial characteristics

| Characteristic (n=365) | | n | Median | IQR | P value |
|---|---|---|---|---|---|
| Setting | Hospital | 212 | 0.90 | 0.4–2.29 | 0.009*† |
| | General practice | 51 | 0.71 | 0.32–1.18 | |
| | Mixed | 56 | 1.01 | 0.47–2.64 | |
| | Community | 29 | 2.44 | 0.62–6.41 | |
| | Other | 17 | 1.89 | 0.76–11.7 | |
| Arms | 2 | 278 | 1.10 | 0.41–2.76 | 0.935‡ |
| | 3 | 55 | 0.85 | 0.45–2.1 | |
| | 4 | 22 | 1.04 | 0.57–1.91 | |
| | >4 | 10 | 0.85 | 0.42–8.85 | |
| Control type | Placebo | 85 | 0.84 | 0.38–1.93 | 0.145§ |
| | Active | 280 | 1.03 | 0.43–3.22 | |
| Original target recruitment | ≤200 | 41 | 1.18 | 0.47–2.65 | 0.008‡ |
| | 201–400 | 93 | 0.78 | 0.36–2.01 | |
| | 401–600 | 84 | 0.84 | 0.43–1.96 | |
| | 601–800 | 40 | 1.13 | 0.46–2.88 | |
| | >800 | 105 | 1.49 | 0.55–4.72 | |
| Final target recruitment | ≤200 | 45 | 0.89 | 0.27–2.55 | <0.001‡ |
| | 201–400 | 103 | 0.76 | 0.34–1.96 | |
| | 401–600 | 83 | 0.86 | 0.44–2.26 | |
| | 601–800 | 41 | 1.17 | 0.57–4.23 | |
| | >800 | 93 | 1.66 | 0.58–5.17 | |
| Total recruitment | ≤200 | 63 | 0.50 | 0.17–1.6 | <0.001‡ |
| | 201–400 | 90 | 0.78 | 0.37–2.07 | |
| | 401–600 | 81 | 1.15 | 0.49–2.41 | |
| | 601–800 | 39 | 1.03 | 0.57–3.85 | |
| | >800 | 92 | 1.96 | 0.68–6.23 | |
| Timing of final follow-up | ≤1 month | 19 | 1.29 | 0.42–2.26 | 0.054‡ |
| | 1< months ≤6 | 82 | 1.14 | 0.38–4.14 | |
| | 6< months ≤18 | 170 | 0.98 | 0.46–2.33 | |
| | >18 months | 85 | 0.71 | 0.36–2.02 | |

*The category 'other' was not included in Kruskal-Wallis test.
†P values are reported from a Kruskal-Wallis test.
‡P values are reported from a Jonckheere-Terpstra test.
§P values are reported from a Mann-Whitney U test.

to 2020. However, there is some evidence of a difference in recruitment rates between the five NIHR journals, and therefore, any improvement may be due to the inclusion of trials from the journals (EME, PGfAR, HS&DR and PHR) that were not included in the review by Walters *et al*.[8] There is still cause for some concern with 30% (118/388) of trials revising their original recruitment target with the majority (67% (79/118)) revising the target downwards, and a third (128/388) of trials having an extension to their recruitment period. These findings remain consistent with the concerns expressed by CTU directors.[13]

The median consent and retention rate are consistent with the result of Walters *et al*.[8] The retention figure may

be an overestimate as it will be affected by trials using time to event outcomes, where missing outcomes are censored at the time of loss to follow-up but included in analyses using survival models. The target sample size for any trial should allow for participant withdrawals and loss to follow-up[40] with the expected withdrawal proportion obtained from reports of studies conducted in the same clinical area.[20] However, if no such information is available then a pragmatic approach would be to use the median retention rate from this review (88%) and assume an expected withdrawal proportion of at least 10%.

This study has the following limitations. First, the review was restricted to publicly funded trials published

**Table 6** Association between the trial retention rate (% of randomised participants with valid primary outcome data for analysis) and trial characteristics

| Characteristic (n=388) | | n | Median | IQR | P value |
|---|---|---|---|---|---|
| Setting | Hospital | 219 | 91.5 | 82.2–97.8 | 0.001*† |
| | General practice | 55 | 84.0 | 76.6–91.3 | |
| | Mixed | 61 | 87.3 | 79.7–97.3 | |
| | Community | 34 | 84.9 | 75.4–90.8 | |
| | Other | 19 | 84.2 | 74.9–96.5 | |
| Arms | 2 | 290 | 89.9 | 81–97.4 | <0.001‡ |
| | 3 | 61 | 84.4 | 72.4–93.6 | |
| | 4 | 24 | 83.2 | 79.6–88.2 | |
| | >4 | 13 | 80.2 | 73.4–96.4 | |
| Control type | Placebo | 87 | 89.8 | 79.1–97.3 | 0.614§ |
| | Active | 301 | 87.8 | 80.3–96.4 | |
| Final target recruitment | ≤200 | 53 | 88.6 | 79.6–96.4 | 0.003‡ |
| | 201–400 | 112 | 86.1 | 77.1–94.1 | |
| | 401–600 | 84 | 86.8 | 78.9–95.7 | |
| | 601–800 | 42 | 84.4 | 80.4–90.9 | |
| | >800 | 97 | 96.3 | 85.3–99.1 | |
| Total recruitment | ≤200 | 72 | 87.9 | 74.5–96.2 | 0.001‡ |
| | 201–400 | 99 | 87.3 | 79.3–94.9 | |
| | 401–600 | 82 | 86.4 | 80.6–94.1 | |
| | 601–800 | 39 | 86.2 | 82.2–91.4 | |
| | >800 | 96 | 95.8 | 82.4–99 | |
| Timing of final follow-up | ≤1 month | 20 | 92.2 | 78.7–99 | 0.518‡ |
| | 1< months ≤6 | 87 | 88.5 | 79.8–96.7 | |
| | 6< months ≤18 | 181 | 88.2 | 79.5–96.4 | |
| | >18 months | 88 | 87.8 | 80–95.5 | |

*The category 'other' was not included in Kruskal-Wallis test.
†P values are reported from a Kruskal-Wallis test.
‡P values are reported from a Jonckheere-Terpstra test.
§P values are reported from a Mann-Whitney U test.

**Table 7** Comparison of the current review with three previous reviews in terms of successful recruitment to target sample size and extensions to recruitment

| Review | McDonald et al[7] | Sully et al[5] | Walters et al[8] | This study | This study |
|---|---|---|---|---|---|
| **Recruitment period** | **1994–2002** | **2002–2008** | **2004–2016** | **2017–2020** | **1997–2020** |
| **No of trials in the study** | **N=122** | **N=73** | **N=151** | **N=174** | **N=388** |
| Recruited 100% of original target | 38 of 122 (31%) | 40 of 73 (55%) | 61 of 151 (40%) | 107 of 174 (61%) | 207 of 388 (53%) |
| Original target was revised | 42 of 122 (34%) | 14 of 73 (19%) | 52 of 151 (34%) | 54 of 174 (31%) | 118 of 388 (30%) |
| Original target revised upward | 6 of 42 (14%) | 5 of 14 (36%) | 11 of 52 (21%) | 23 of 54 (43%) | 39 of 118 (33%) |
| Original target revised downward | 36 of 42 (86%) | 9 of 14 (64%) | 41 of 52 (79%) | 31 of 54 (57%) | 79 of 118 (67%) |
| Recruited 80% of original target | 67 of 122 (55%) | 57 of 73 (78%) | 95 of 151 (63%) | 139 of 174 (80%) | 288 of 388 (74%) |
| Recruited 100% of revised target | 19 of 42 (45%) | 10 of 14 (71%) | 28 of 52 (54%) | 35 of 54 (65%) | 80 of 118 (68%) |
| Recruited 80% of revised target | 34 of 42 (80%) | 13 of 14 (93%) | 48 of 52 (92%) | 48 of 54 (89%) | 107 of 118 (91%) |
| Extended their recruitment | 65 of 122 (54%) | 33 of 73 (45%) | 49 of 151 (32%) | 65 of 174 (37%) | 128 of 388 (33%) |

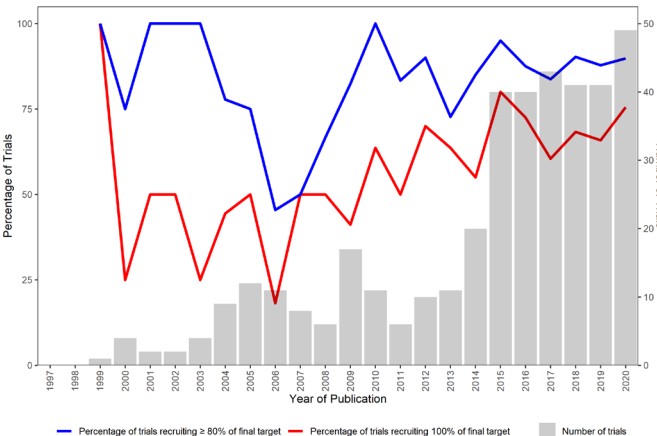

**Figure 3** Number of trials and percentage of trials recruiting 100% and ≥80% of the final sample size target from 1997 to 2020

in the NIHR Journals Library, which may limit the generalisability of the findings. It is possible that that problems with recruitment and retention of participants in NHIR funded trials will be less pronounced than in other trials due to the rigorous appraisal of feasibility prior to funding and the ongoing monitoring during the conduct of the trial. However, as the NIHR Journals Library intends to publish all research from EME, HS&DR, HTA, PGfAR and PHR funded projects, it has less chance of publication bias compared with a review of other journals where publishing is more selective and information related to recruitment is published in less detail. Second, the data extraction was conducted by several independent reviewers and although reviewers conferred to try and ensure consistency and quality assurance checks were completed on a sample of reports, it is possible that errors have occurred. Third, the calculation of recruitment rates was limited by the information reported. For some trials, centre-specific recruitment information was not available meaning that crude recruitment rates, assuming all centres were recruiting for the same time period, were calculated. In these cases, the calculated recruitment rate may be an underestimate of the true recruitment rate.

This review found considerable variation in the consent, recruitment and retention rates in publicly funded RCTs. Although the majority of (6 out of 10) trials in this review achieved their final target sample; 3 out of 10 trials published in NIHR Journals Library revised their original target sample size (downwards in 7 out of 10 trials). Investigators should bear this in mind at the planning stage of their study and not be overly optimistic about their recruitment projections.

**Contributors** RMJ is the guarantor of the study, had full access to all the data in the study, and is responsible for the integrity of the data and the accuracy of the data analysis. RMJ and SJW contributed to the study concept and design. RMJ, RA, JH, AR and IS contributed to the selection of data and conducted the data extraction. RMJ conducted the data analysis and drafted the manuscript. RMJ, RMS and SJW contributed to the quality assurance check of the data. All authors critically revised the manuscript and approved the final manuscript.

**Funding** SJW is an NIHR Senior Investigator supported by the NIHR (NF-SI-0617-10012) for this research project.

**Disclaimer** The views expressed in this publication are those of the authors and not necessarily those of the NIHR, NHS or the UK Department of Health and Social Care.

**Competing interests** RMJ, RMS and SJW received funding across various projects from the National Institute for Health Research (NIHR).

**Patient and public involvement** Patients and/or the public were not involved in the design, or conduct, or reporting, or dissemination plans of this research.

**Patient consent for publication** Not applicable.

**Ethics approval** This study does not involve human participants.

**Provenance and peer review** Not commissioned; externally peer reviewed.

**Data availability statement** Data are available on reasonable request. The information extracted in this review is based on published trials in the NIHR Journals Library. The extracted data is available on reasonable request from the corresponding author at r.jacques@sheffield.ac.uk.

**ORCID iDs**
Richard M Jacques http://orcid.org/0000-0001-6710-5403
Rebecca M Simpson http://orcid.org/0000-0003-1677-5938
Stephen J Walters http://orcid.org/0000-0001-9000-8126

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
