## [Reviewer comments · BMJ Open]

ARTICLE DETAILS

TITLE (PROVISIONAL)	Recruitment, consent and retention of participants in randomised controlled trials: a review of trials published in the National Institute for Health Research (NIHR) Journals Library (1997 – 2020)
AUTHORS	Jacques, Richard; Ahmed, Rashida; Harper, James; Ranjan, Adya; Saeed, Isra; Simpson, Rebecca; Walters, Stephen

VERSION 1 – REVIEW

REVIEWER	Prescott, Robin University of Edinburgh, Medical Statistics Unit
REVIEW RETURNED	03-Dec-2021

GENERAL COMMENTS	The authors have produced a very competent review of recruitment and retention of participants in NIHT published RCTs from 1997 to 2020. The methodology of the review is sound, and the presentation is both clear and appropriate. There are very few comments that I can offer to try to improve this paper. The one area where I feel there could be greater clarity is the handling of pilot trials. Within the section on Inclusion/Exclusion Criteria, there is a sentence that states “Reports of internal pilot trials that either went on to a full trial or were terminated due to recruitment issues were included in the review”. Within the Results section it is identified that 95 pilot/feasibility RCTs were excluded. The distinction between these two classes of pilot trials could be made clearer in my view. I would also like to see the rationale explained within the paper for the decisions with respect to inclusion or exclusion. I interpret that the inclusion of an internal pilot trial that was included in the review would be handled as part of the full trial, but that slight ambiguity could be usefully clarified. I would like to add a comment on the representativeness of the trials funded by NIHR. These undergo a rigorous appraisal of grant applications in which trial feasibility is assessed, and there is great competition for funding. There is also ongoing monitoring of studies by NIHR, to try to help with any difficulties that might emerge during the conduct of the trial. It is therefore plausible that the problems with recruitment and retention of participants in these trials will be less pronounced than in most trials. It is a consideration that reinforces the cautionary message in the Conclusions of this review.
--

REVIEWER	Gillies, Katie University of Aberdeen, Health Services Research Unit
-----------------	---

REVIEW RETURNED	08-Dec-2021
GENERAL COMMENTS	"Recruitment and retention of participants in randomised controlled trials: a review of trials published in the National Institute for Health Research (NIHR) Journals Library (1997 – 2020)." Jacques et al Thank you for the opportunity to review this manuscript which presents the findings of an updated review of trials published in the NIHR Journals to explore reporting of recruitment and retention of participants. Overall this is a really well written and reported manuscript and the findings contribute updated data to evidence base in this area. I have some general consideration and then some specific points for consideration, all of which are relatively minor. These considerations are listed below. Abstract  • The authors state the objective was to review 'the consent, recruitment and retention rates' – the paper goes on to report recruitment and retention but not consent. The conclusion section also restates consent as having considerable variation but I'm not sure the results presented provide that evidence. Methods  • Page 6 – Trial identification. States NIOHR is largest funder of health and social care research in England – should this be the UK? • Page 9 – Analysis: as per earlier comment , the information on consent rate calculation states that this was calculated from 'the percentage of eligible participants consented and randomised'. How is this different to the recruitment rate defined on page 8 as 'the number of participants recruited and randomised'? How does 'recruited' differ to 'consented'? • Page 9 – were retention rates taken directly from the NIHR reports directly with no further adjustment in your analyses? Please specify as such. Results  • Page 11, first para – Please could the authors add in the sample size of the (obstetrics_ trial that they report as having the highest number recruited [ref 22]. • Page 11, first para – It is reported that 63% of trials recruited to their final target - was this on time and on budget? Discussion  • Page 13, para 2 – Could the authors comment on why they believe recruitment has improved in the cohort of trials in 2017-2020 now included in this updated review. • I also wondered if the authors can comment on whether the trials across the different NIHR programmes differ by recruitment or retention and whether this could also account for a change over the time frame if the NIHR portfolio overall has changed. • Page 13, para 3 – last sentence stating that 10% being a pragmatic approach to attrition needs referenced. • Page 14, para 2 – findings again highlight variation in consent. This may need addressed based on earlier comments.
REVIEWER	Pontoppidan, Maiken Nationale Forskningscenter for Velfard, Child and Family
REVIEW RETURNED	10-Dec-2021

GENERAL COMMENTS	This is a very relevant and great paper! Congratulations! I only have a few minor comments:  1. maybe you want to add this website with UK guidelines for recruitment? http://www.hra-decisiontools.org.uk/consent/ 2. the included papers are published until 31st December 2020. It would be great if you can include papers from 2021. Usually, the search has to be updated when it is 12 months old. 2. inclusion criteria - is there any definition on what areas of trials can be included? 3. I am not a statistics expert so cannot review the appropriateness of the statistical method used 4. it seems that single-center trials recruit at a much higher monthly rate than multicenter trials. can you comment on this in the discussion - why is this?
--

VERSION 1 – AUTHOR RESPONSE

Reviewer 1

The one area where I feel there could be greater clarity is the handling of pilot trials. Within the section on Inclusion/Exclusion Criteria, there is a sentence that states “Reports of internal pilot trials that either went on to a full trial or were terminated due to recruitment issues were included in the review”. Within the Results section it is identified that 95 pilot/feasibility RCTs were excluded. The distinction between these two classes of pilot trials could be made clearer in my view. I would also like to see the rationale explained within the paper for the decisions with respect to inclusion or exclusion. I interpret that the inclusion of an internal pilot trial that was included in the review would be handled as part of the full trial, but that slight ambiguity could be usefully clarified.

We agree that some further clarification around pilot/feasibility trials would be useful. External pilot trials were excluded because they do not contribute outcome data to the main trial and instead are used to estimate parameters in the design of the main trial. The description of the inclusion/exclusion criteria have been amended to reflect this.

“Reports of external pilot/feasibility studies were excluded as they do not contribute outcome data to the main trial and are instead often used to estimate parameters such as the number of eligible participants, willingness of participants to be randomised and follow-up rates needed for the design of the main study. Reports of internal pilot trials that either went on to contribute outcome data to a full trial or were terminated prior to the full trial because of recruitment issues were included in the review.”

We have also amended the flow diagram in Figure 1 to show that the excluded studies are external pilot/feasibility RCTs.

I would like to add a comment on the representativeness of the trials funded by NIHR. These undergo a rigorous appraisal of grant applications in which trial feasibility is assessed, and there is great competition for funding. There is also ongoing monitoring of studies by NIHR, to try to help with any difficulties that might emerge during the conduct of the trial. It is therefore plausible that the problems with recruitment and retention of participants in these trials will be less pronounced than in most trials. It is a consideration that reinforces the cautionary message in the Conclusions of this review.

We agree that the NIHR grant application process and ongoing monitoring of trials could impact on the generalisability of these findings. We have expanded the limitations section to include this: “It is possible that that problems with recruitment and retention of participants in NHIR funded trials will be less pronounced than in other trials due to the rigorous appraisal of feasibility prior to funding and the ongoing monitoring during the conduct of the trial.”

Reviewer 2

Abstract

- The authors state the objective was to review ‘the consent, recruitment and retention rates’ – the paper goes on to report recruitment and retention but not consent. The conclusion section also restates consent as having considerable variation but I’m not sure the results presented provide that evidence.

Methods

The median (IQR) consent rate (percentage of eligible participants who consented and were randomised is reported in Table 4). This was originally omitted from the abstract but we agree that as a secondary outcome for the review it should be included. We have updated the primary and secondary outcome measures and results sections of the abstract to reflect this. We have also updated the title and row descriptions for Table 4 to make it clear that it is reporting the consent, recruitment and retention rates.

- Page 6 – Trial identification. States NIHR is largest funder of health and social care research in England – should this be the UK?

The 2019/20 NIHR Annual Report states that “The NIHR is the largest funder of health and care research in England”. We have therefore not changed this sentence to say UK but have added the reference to the NIHR Report.

- Page 9 – Analysis: as per earlier comment, the information on consent rate calculation states that this was calculated from ‘the percentage of eligible participants consented and randomised’. How is this different to the recruitment rate defined on page 8 as ‘the number of participants recruited and randomised’? How does ‘recruited’ differ to ‘consented’?

The recruitment rate describes how many participants are recruited and randomised and per centre per month. The consent rate is the percentage of eligible participants that consented and were randomised. We have added paragraph to the analysis section describing the calculation of secondary outcomes.

“The secondary outcomes for the review were the target sample size and whether it was achieved, the consent rate and the retention rate. The consent rate was calculated as the percentage of eligible participants that consented and were randomised (i.e. the total number of participants recruited and randomised divided by the number of eligible participants). The retention rate was calculated as the percentage of randomised participants that were assessed for the primary outcome and included in the analysis of the primary outcome (i.e. the number of participants included in the analysis of the primary outcome divided by the number of participants recruited and randomised).”

- Page 9 – were retention rates taken directly from the NIHR reports directly with no further adjustment in your analyses? Please specify as such.

The retention rates were calculated using information reported in the CONSORT flow diagrams of trials and/or information from the tables reporting the primary analysis. We have added a paragraph to the analysis section describing the calculation of the secondary outcomes (see comment above).

Results

- Page 11, first para – Please could the authors add in the sample size of the (obstetrics_ trial that they report as having the highest number recruited [ref 22].

The target sample size for this trial was 2232. We have added this to the text.

“The highest recruitment rate (57.75 participants per centre per month) was in a trial comparing medical to surgical termination of pregnancy with a target sample size of 2232 women”

- Page 11, first para – It is reported that 63% of trials recruited to their final target - was this on time and on budget?

We have not collected information on the trial budgets but the text in this paragraph has been updated to indicate the number of trials that reached their final recruitment target but required an extension to the recruitment time period.

“Overall, 63% (245/388) of trials recruited to their final recruitment target but 32% (79/245) of these trials required an extension to their recruitment period to meet the target. A further 22% (86/388) of trials recruited to within 80% of their final recruitment target with 36% (31/86) of these trials having an extension to their recruitment period”

Discussion

- Page 13, para 2 – Could the authors comment on why they believe recruitment has improved in the cohort of trials in 2017-2020 now included in this updated review.

It is possible that investigators are being more cautious when specifying the recruitment period and/or the NIHR is making investigators be more realistic through the grant application and review process. However, we haven't commented on this as this is just speculation on our behalf and we do not have any evidence to support it.

- I also wondered if the authors can comment on whether the trials across the different NIHR programmes different by recruitment or retention and whether this could also account for a change over the time frame if the NIHR portfolio overall has changed.

We have updated the results in Table 4 to show the consent, recruitment and retention rates by journal with a description of these results in the “Recruitment, Consent and Retention Rates” section. There is some variability between the journals in terms of recruitment rates with PHR having a higher median rate than the over four journals (although this is only based on 6 trials).

We have also added a sentence to the Discussion and Conclusions section highlighting that any improvements may be due to the inclusion of additional journals not included in the previous review.

“However, there is some evidence of a difference in recruitment rates between the five NIHR journals and therefore any improvement may be due to the inclusion of trials from the journals (EME, PGfAR, HS&DR and PHR) that were not included in the review by Walters et al.(8)”

- Page 13, para 3 – last sentence stating that 10% being a pragmatic approach to attrition needs referenced.

This sentence is our opinion based on the median retention rate being 88%. The sentence has been amended to make this clear.

“However, if no such information is available then a pragmatic approach would be to use the median retention rate from this review (88%) and assume an expected withdrawal proportion of at least 10%.”

- Page 14, para 2 – findings again highlight variation in consent. This may need addressed based on earlier comments.

We have added the median consent rates to the discussion and conclusions section.

VERSION 2 – REVIEW

REVIEWER	Prescott, Robin University of Edinburgh, Medical Statistics Unit
REVIEW RETURNED	17-Jan-2022

GENERAL COMMENTS	I congratulate the authors on a very nice presentation. I have no further comments.
---

REVIEWER	Gillies, Katie University of Aberdeen, Health Services Research Unit
REVIEW RETURNED	12-Jan-2022

GENERAL COMMENTS	Thankyou to the authors for addressing my comments.
---